# AI-Supported Comprehensive Detection and Quantification of Biomarkers of Subclinical Widespread Diseases at Chest CT for Preventive Medicine

**DOI:** 10.3390/healthcare10112166

**Published:** 2022-10-29

**Authors:** Viktoria Palm, Tobias Norajitra, Oyunbileg von Stackelberg, Claus P. Heussel, Stephan Skornitzke, Oliver Weinheimer, Taisiya Kopytova, Andre Klein, Silvia D. Almeida, Michael Baumgartner, Dimitrios Bounias, Jonas Scherer, Klaus Kades, Hanno Gao, Paul Jäger, Marco Nolden, Elizabeth Tong, Kira Eckl, Johanna Nattenmüller, Tobias Nonnenmacher, Omar Naas, Julia Reuter, Arved Bischoff, Jonas Kroschke, Fabian Rengier, Kai Schlamp, Manuel Debic, Hans-Ulrich Kauczor, Klaus Maier-Hein, Mark O. Wielpütz

**Affiliations:** 1Department of Diagnostic and Interventional Radiology, Subdivision of Pulmonary Imaging, University Hospital of Heidelberg, Im Neuenheimer Feld 420, 69120 Heidelberg, Germany; 2Translational Lung Research Center Heidelberg (TLRC), German Center for Lung Research (DZL), Im Neuenheimer Feld 156, 69120 Heidelberg, Germany; 3Department of Diagnostic and Interventional Radiology with Nuclear Medicine, Thoraxklinik at the University Hospital of Heidelberg, Röntgenstr. 1, 69126 Heidelberg, Germany; 4Division of Medical Imaging Computing, German Cancer Research Center Heidelberg, Im Neuenheimer Feld 223, 69120 Heidelberg, Germany; 5Pattern Analysis and Learning Group, Department of Radiation Oncology, University Hospital of Heidelberg, Im Neuenheimer Feld 672, 69120 Heidelberg, Germany; 6Medical Faculty, University of Heidelberg, Im Neuenheimer Feld 672, 69120 Heidelberg, Germany; 7Interactive Machine Learning Research Group, German Cancer Research Center Heidelberg, Im Neuenheimer Feld 223, 69120 Heidelberg, Germany; 8Department of Diagnostic and Interventional Radiology, Medical Center, Faculty of Medicine Freiburg, University of Freiburg, Hugstetter Str. 55, 79106 Freiburg, Germany

**Keywords:** artificial intelligence, comorbidities, machine learning, computer assisted image analysis, medical image processing, medical computing, CT imaging postprocessing, preventive medicine, radiomics

## Abstract

Automated image analysis plays an increasing role in radiology in detecting and quantifying image features outside of the perception of human eyes. Common AI-based approaches address a single medical problem, although patients often present with multiple interacting, frequently subclinical medical conditions. A holistic imaging diagnostics tool based on artificial intelligence (AI) has the potential of providing an overview of multi-system comorbidities within a single workflow. An interdisciplinary, multicentric team of medical experts and computer scientists designed a pipeline, comprising AI-based tools for the automated detection, quantification and characterization of the most common pulmonary, metabolic, cardiovascular and musculoskeletal comorbidities in chest computed tomography (CT). To provide a comprehensive evaluation of each patient, a multidimensional workflow was established with algorithms operating synchronously on a decentralized Joined Imaging Platform (JIP). The results of each patient are transferred to a dedicated database and summarized as a structured report with reference to available reference values and annotated sample images of detected pathologies. Hence, this tool allows for the comprehensive, large-scale analysis of imaging-biomarkers of comorbidities in chest CT, first in science and then in clinical routine. Moreover, this tool accommodates the quantitative analysis and classification of each pathology, providing integral diagnostic and prognostic value, and subsequently leading to improved preventive patient care and further possibilities for future studies.

## 1. Introduction

Each patient is a complex individual with different, subtle and often specialty-overlapping medical conditions, or even hitherto unrevealed subclinical comorbidities frequently associated with one another [1,2].

The WHO constitutes obesity as the largest global chronic health problem in adults worldwide, with 21.5% of men and 24.5% of women of the European population affected, oftentimes resulting in a complex systemic metabolic syndrome [3]. The Framingham Heart Study has shown that cardiovascular diseases, which are known to be associated with metabolic syndrome are the leading causes of death globally [4].

Smoking is one of the most important risk factors for COPD (Chronic obstructive pulmonary disease) affecting about 10.2–26.1% of the population, leading to progressive lung parenchyma destruction. Hence, it is the third leading cause of death worldwide [5,6]. Additionally, approximately 51% of smokers above the age of 50 have intrapulmonary nodules [7]. These incidental findings carry the risk of malignancy classified under the Fleischner and LungRADS criteria, based on nodule size and density [8]. Moreover, systemic bone mineral density (BMD) loss, osteopenia and osteoporosis, show strong associations with COPD and affect about 500 mio. People worldwide, oftentimes leading to lifelong physical disability caused by spontaneous fractures and further worsening outcomes in combination with coexisting comorbidities [9,10].

A computed tomography (CT) scan of the ’hest Is usually acquired for specific diagnostic indications, e.g., suspicion of a pulmonary nodule, with focused reporting. Nevertheless, the images contain information beyond these tasks, such as the cardiovascular system, metabolic disorders and musculoskeletal pathologies. A holistic report may synthesize these findings connected to otherwise neglected comorbidities together with available clinical data to perform a sufficient comprehensive risk assessment (often termed ‘opportunistic screening’) [11,12]. In particular, secondary diagnoses and comorbidities play a significant role in overall prognosis and disease management of multiple conditions, and may further be used for preventive medicine and life-style interventions [1,2,13,14,15,16,17,18].

Although the importance of comorbidities is well known, one of the key difficulties is detecting and quantifying these comorbidities on imaging, because subtle manifestations are oftentimes beyond discriminatory capabilities of the human visual system. Moreover, subjective visual image reading is inherently qualitative, while a risk assessment, e.g., of bone mineral density or in evaluation of coronary calcification is based on quantitative data [19,20]. Precise quantification may require specific imaging preconditions, multiple, time-consuming measurements and sometimes additional image postprocessing software for each item.

Studies have shown the advanced abilities of automated artificial intelligence (AI)-based imaging analysis in Radiology, especially for two reasons: complex image analyses can be performed user-independently within second-short timeframes and were shown to perform up to 11.5% percent better than radiologists, e.g., in breast cancer prediction with an AUC of 0.889 (95% CI 0.871, 0.907; n = 25,856 patients) [21,22,23,24]. Similar to increasing sub-specialization in medicine, available computer-based image analysis tools are focused on specified tasks, e.g., typically “emphysema”, “pulmonary embolism”, “nodule”, “aortic dissection” or more recently, “COVID-19 pneumonia” giving a precise answer to each question in disregard of coexisting pathologies [25,26,27,28,29].

To facilitate large-scale studies of imaging biomarkers of comorbidities, and to provide a holistic overview of an individual, the aim of the Comorbidome study was to create a multidimensional AI-based tool with synchronized automated detection and quantification of the most common pulmonary, metabolic, cardio-vascular and musculoskeletal comorbidities in chest CT scans. Moreover, the AI workflow includes an automated summery of the results in a structured report with reference to normal values including guideline-based recommendations for treatment, additional diagnostic work-/or follow-up, as well as sample images of detected pathologies. The following report is intended to display the concept of our prototype software and contains work in progress with anticipation of additional detailed evaluation and validation of each algorithm.

## 2. Materials and Methods

### 2.1. Team

For Comorbidome a multidisciplinary, translational team of radiologists, computer scientists, physicists, mathematicians and radiographers from the University Hospital Heidelberg (UKHD), the Thoraxklinik Heidelberg, the German Cancer Research Center (DKFZ) and Translational Lung Research Center (TLRC) at the German Center for Lung Research (DZL) combined expertise in each field with the aim to build a subspecialty-spanning AI-tool for analyzing comorbidities on chest CT and to implement the technical interface, infrastructure and process pipeline for clinical routine and further scientific projects.

### 2.2. Training Data

This multicentric study was conducted in accordance with the Declaration of Helsinki, and approved by the Institutional Review Board of the Heidelberg medical faculty (S-937/2020). A total of >10,000 CT datasets and corresponding data from the clinical documentation system were available for five years retrospectively (1 January 2016–31 December 2020). Varying acquisition parameter resulting from different clinical indications on seven different CT scanners at the University Hospital Heidelberg and Thoraxklinik Heidelberg ensured a heterogeneous training dataset.

### 2.3. Algorithm Training

For some tasks, deep learning models were pre-trained on publicly available datasets and optimized for their subsequent use. Otherwise, models were newly trained on selected clinical CT datasets from UKHD. Therefore, ground truth segmentations were created manually and semi-automatically by experienced radiologists. Labelling all required regions of interest in CT-image data was performed using the medical imaging interaction toolkit (MITK, 2 April 2018, DKFZ) [30]. This training-process was supported in some cases by pre-segmentations from clinically available software tools (syngo.via, VB60A_HF02, Siemens Healthineers, Erlangen; IntelliSpace Portal, 12.1.2.11402, Philips, Amsterdam; YACTA v2.9.3.7. by Oliver Weinheimer) [31,32,33,34,35,36]. Thereafter, further Digital Imaging and Communications in Medicine (DICOM) data were analyzed by the trained models and results were revised and corrected by experienced radiologists. The corrected segmentations were then used for algorithm re-training to obtain optimized task specific models. All deep segmentation models were trained using nnU-Net which is the first fully automated neural network configuration, optimization and training framework for biomedical image segmentation [37,38]. It offers 2D, 3D and multi-stage 3D Cascaded U-Net-based convolutional-deconvolution neural network architectures. Specifically, it employs skip connections, strided and transposed convolutions for down- and up-sampling, instance normalization and LeakyRELU in each block, additional deep supervision, stochastic gradient descent optimization and a sum of cross-entropy and Dice loss as a loss function. In its core, nnU-Net uniquely introduces a self-configuration framework for data pre-processing, network architecture setup, model training, segmentation postprocessing and optimized model selection for any task, thereby even surpassing most handcrafted approaches for supervised segmentation model training. Its generic design enables state-of-the-art performance for biomedical image segmentation, without employing manual intervention.

As a part of this, preprocessing of image data was determined and applied automatically. For CT data as encountered in the presented settings, the image data undergo a clipping to [0.5, 99.5] percentiles of image intensities as encountered in the segmented regions of the underlying training set, and a z-score normalization based on mean and standard deviation of all intensities. Furthermore, all patients are resampled to the median voxel spacing of the underlying dataset based on third order spline interpolation. Input patches are augmented using random rotations, scaling, elastic deformations, gamma correction and mirroring by default.

Based on this, each subgroup with expertise in a certain specialty independently addressed specific clinical and technical tasks, as detailed subsequently.

#### 2.3.1. Adipose Tissue and Metabolic Imaging

The composition of visceral and subcutaneous abdominal adipose tissue as a risk factor for cardiovascular events and a metabolic marker in cancer has been quantified together with hepatic steatosis [3,39,40,41]. Previous approaches have performed automated segmentation of abdominal body composition based on thresholding and region growing, in conjunction with probabilistic filtering of intensities and with active contours to increase robustness [42,43]. Recent approaches have proposed modern data-driven approaches based on trained 2D- and 3D-CNNs [44,45]. In our setting, to benefit from the performance of a well-optimized encoder-decoder architecture while reducing manual annotation effort, a nnU-Net based algorithm was trained to automatically segment body compartments in a 2D slice at the lumbar spine level of L2/L3. For training, 2D slices were extracted from in-house CT volumes of 25 selected patients at UKHD, at slightly varying L2/L3 lumbar spine levels in order to reflect expected slice detection and extraction variabilities during inference. Ground truth annotations were provided by an expert radiologist, using a threshold-based pre-segmentation followed by correction via manual delineation. The fat area within a compartment was measured after post-processing with a binary threshold filter for fat (−190 HU to −30 HU). Population-based reference values for comparison were depicted from the analysis of the Framingham Heart Study [46]. Liver fat content measurement was performed using an automated, ball-shaped region of interest in the right liver lobe in unenhanced CT images. Automated placement depends on correct lung segmentation. Currently, an algorithm from Irlbeck et al. is used for this purpose [46]. Reference values for grading steatosis were taken from (Table 1) [47].

#### 2.3.2. Cardiovascular Comorbidities

For cardiovascular risk stratification, the identification, quantification, density assessment and scoring of coronary artery and aortic calcifications were implemented for both non-contrast and contrast-enhanced CT scans of chest and abdomen (Table 1) [4,48,49,50,51]. Quantification includes computation of the well-validated Agatston Score [4,48,52,53]. Previous works have addressed automated segmentation and quantification of calcification using various machine learning approaches, ranging from random forests [54,55] to single-stage and multi-stage deep learning approaches [56,57]. For the sake of potentially increased robustness from a multi-stage approach, a two step-based, supervised nnU-Net training was performed, aiming for accurate artery calcium detection separated by vascular compartments. Subsequently, calcium segmentation, a labeling segmentation prevented false positive findings caused by adjacent bone or dense foreign material, such as intravascular catheters.

#### 2.3.3. Pulmonal Parenchyma Abnormalities

A previously well-validated, in-inhouse developed software YACTA (Yet Another CT Analyzer) has been integrated into the workflow for profound multidimensional analysis of lung density, emphysema, large and small airway disease (Table 1) [31,32,33,34,37,58,59,60,61].

#### 2.3.4. Pulmonary Nodules

Building upon previous efforts of computational algorithms for pulmonary nodule detection, [62] a state-of-the-art deep-learning-based approach for lung nodule detection was adapted and integrated (nnDetection) [38,63,64]. Analogously to nnU-Net, this approach aims at generic high-performance for medical detection tasks, by automating design choices and training optimizations based on dataset properties and built-in heuristics. Competitiveness was proven in the LUNA16 (Lung Nodule Analysis 2016) nodule detection challenge, where nnDetection outperformed almost all other state-of-the-art approaches [65,66,67,68]. The method was trained on 1035 CTs from the LIDC (Lung Image Database Consortium) dataset [69], using an assembling from subsets in a 5-fold cross validation. Automated measurement of each nodule was performed providing risk classification by probability and LungRADS (Lung CT Screening Reporting & Data System) (Table 1) [70,71]. The nnDetection is a self-configuring medical object detection method which was developed and validated on a pool of 10 datasets including focal pathologies such as lung nodules, rib fractures and aneurysms. Using fixed, rule-based and empirical parameters, it adapts itself to new datasets without manual intervention. Resulting models were ensembled during inference. In order to extract volume measurements complementing the bounding box-based diameter measurements, a segmentation output was additionally derived from nnDetection.

#### 2.3.5. Osteoporosis

Modern approaches for automated vertebra segmentation have been proposed that strongly rely on CNN-based deep learning approaches. U-Net-architectures were employed early on [72], followed by more complex multi-stage approaches that have shown to be advantageous [73,74,75]. A two-step deep learning approach was implemented which first performs a labeling of the individual vertebral bodies and then extracts the trabecular bone for measurements of BMD. The corresponding training for vertebral body segmentation was performed on 374 multidetector, multi-site, publicly available CT spine dataset from the VerSe (vertebrae segmentation) challenges 2019 and 2020. The cortical and trabecular bone extraction model was trained on manually refined segmentations of an in-house dataset, based on pre-segmentations derived by the vertebral body segmentation model [37,76]. The measured densities in trabecular bone are converted to BMD based on in-house acquired, phantom based standard reference values, allowing for T-score evaluation and thereby providing a comparability to the clinical gold standard.

### 2.4. Technical Set-Up

The Joint Imaging Platform (JIP) served as the technical basis for this project. The JIP was designed and created as a joint DKFZ-DKTK (Deutsches Konsortium für Translationale Krebsforschung) cooperation and is based on the Kaapana open-source toolkit [77]. In previous efforts, this technical design had been installed and established at several medical institutions nationwide, providing a uniform platform for developing advanced medical imaging analysis [21,22,27]. Kaapana incorporates several core components, including dcm4chee, Elasticsearch and Kibana for storing, searching and visualizing DICOM data; Kubernetes and Docker for handling containers, Airflow for workflow management and Keycloak for user authentication. The current, deployed Kaapana version is v0.1.3 [77].

The in-house developed and established data transfer software ADiT (Automated DICOM Transfer) ensures a dynamic and continuous, interactive data exchange between the picture archiving and communications systems (PACS) and the JIP and thereby provides the possibility for a data analysis from a multicenter clinical workflow [78]. Additionally, open-source data were used where accessible.

**Table 1 healthcare-10-02166-t001:** Biomarkers of comorbidities for comprehensive quantitative and qualitative analysis of chest CT.

Comorbidity	Parameter
Adipose tissue and metabolism	− Visceral fat area (VFA) − Subcutaneous fat area (SFA) − Intermuscular fat area (IMAT) − Liver attenuation − Liver fat content − Fat areas (cm^2^) [46] − Liver fat content (HU) [47,79] − Grading steatosis [79]
Cardiovascular	− Coronary artery calcium (CAC) − Thoracic aortic calcium (TAC) − Abdominal aortic calcium (AAC) − Coronary artery Calcium Scoring (Agatston method) [49,52,53,80,81,82,83]
Pulmonary parenchyma and COPD	− Wall Percentage (%) [32] − AWTPi10 (cm) [84] − Lung Volume (cm^3^) [85] − MLD (HU) [85] COPD phenotyping (PRM %) [58] − Emphysema index (%) [85] − Bronchiectasis index (%) [59] − Ground glass opacity Index (%) [86] − Fibrosis index (%) [86]
Pulmonary nodule	− Number − Volume (mm^3^) − Diameter (x, y, z) − Probability (%) − LungRADS [87]
Osteoporosis	− Mean density (HU) − Hydroxyapatite in trabecular bone (mg/cm^3^) − Bone mineral density (t-score) [88,89,90,91]

## 3. Results

A multidimensional workflow combines different algorithms for a synchronized assessment of the different comorbidities on the JIP server. An automatically assembled structured report then comprises all findings including the quantitative results and displays image thumbnails of detected pathologies.

### 3.1. Workflow

Each algorithm consists of multiple docker containers with connected input and output folders that are integrated in the JIP (Figure 1a) [77]. The containers reflect the subparts of each underlying algorithm pipeline and have been built for the analytical steps in our project, comprising data conversion, preprocessing, segmentation, detection, postprocessing, parameter and feature extraction, result conversion, visualization and automated radiological report generation. Containers are organized in directed acyclic graph structures (DAG) to form predefined workflows, which are run by the workflow automation component of the JIP (JIP Flow). Based on Airflow, the infrastructure allows concurrent execution of different algorithms on incoming datasets while scaling with underlying hardware, thus being capable of analyzing various patients and comorbidities simultaneously (Figure 1b). Furthermore, possibilities for ongoing algorithm development and prospective integration of additional AI-solutions are provided by the platform.

### 3.2. Output

All findings resulting from the automated assessment are displayed in a structured “hAIco” (Heidelberg AI-based detection of Comorbidities) report (Appendix A). Along with the detailed quantitative information on findings, the report includes reference values if available and recommendations for additional diagnostic workup. According to the significance, findings are labelled by colors which is intended to support an intuitive report for physicians and patients. Findings with measured values beyond normal are marked in red. Green labelled results are within range. Values in between these two categories are marked yellow. Cross-sectional images are attached at the end of each report with demonstration of the organ segmentations and measurements, as well as highlighting the pathologic findings (Figure 2 and Figure 3) and the online Appendix A demonstrates the results of the comorbidity analysis of chest CT scans of two patients.

#### 3.2.1. Example No. 1 ‘TAVI Patient’

The algorithm-based, automated analysis of a contrast-enhanced thoraco-abdominal CT scan from a 75-year-old male patient, for the feasibility assessment of a Transcatheter Aortic Valve Implantation (TAVI) showed abnormal values of VFA, an indication for nutrition adjustment (Figure 2, Table 2, Appendix A). In addition to elevated TAG and AAC levels, a high artery calcium deposition was scored with the Agatson method at >300, recommending a high-intensity statin therapy [53,81,82,92]. Furthermore, an incidental pulmonary nodule was detected and classified as Lung-RADS grade 2, suggesting follow-up in 12 months with CT [87]. Bone mineral density acquisition showed pathologically reduced t-scores in need for additional diagnostic workup [19].

#### 3.2.2. Example No. 2 ‘COPD Patient’

The automated evaluation of an un-enhanced chest CT scan from a 66-year-old male patient for the diagnostic workup of chronic lung disease quantified moderate parenchymal lung changes with mild signs of Emphysema, COPD and bronchial wall thickening (Figure 3, Table 3, Appendix A). Additionally, this patient demonstrated an intermediate level of hepatic steatosis hepatis with an otherwise average body fat distribution. With an Agatson score >200 of CAC, this patient is expected to have a high risk of cardiac events. Hence, statin therapy is highly recommended [53,81,82,92]. BMD was significantly reduced with t-scores at osteoporotic level in need of diagnostic workup [19].

## 4. Discussion

With the Comorbidome project, we aim to combine several algorithms to assess the most important comorbidities of diseases of the chest in a single, multidimensional diagnostic workflow. We hereby want to provide a comprehensive tool for opportunistic screening for the most prevalent common diseases captured by routine chest CT with and without contrast media. From the individual perspective, detection of subclinical conditions at asymptomatic stages may offer the possibility of intervention by lifestyle-adaptation or early therapy. On the other hand, urgent findings can be indicated, e.g., coronary artery disease, based on accurate quantification and classification. By analyzing the individual constellation of different comorbidities and by providing recommendations a more comprehensive medical picture of each patient is achieved, contributing to individualized therapeutic management.

In addition to the clinical aspects, this comprehensive approach offers new scientific possibilities in evaluating still unknown correlations of different comorbidities. For example, it has been shown that Osteoporosis is a significant factor for outcome prognosis in atherosclerosis, although the cause of this relation is still unknown and may be addressed by this tool in future [93,94,95]. This technical development may further support the attempt to explain the correlation of osteopenia, fracture risk and sarcopenia, which is diversely discussed in latest studies [96,97,98]. Overall, this infrastructure provides new opportunities, not mere interpretation of single studies, for receiving a holistic picture in epidemiologic large-scale studies, supporting the evaluation of the cause and consequences of the correlations of several diseases, including, e.g., gender, race, economic status and population sub-cohorts.

Since this technical development provides a unique possibility to go beyond evaluation of single diseases, it opens doors for re-evaluation of pre-existing data in a more holistic context. Hence, previously acquired CT images from former studies such as the COSYCONET study with COPD patients or the lung cancer screening study LUSI (Lung cancer Screening Intervention) are planned to undergo additional, retrospective image assessment to re-evaluate the existing data for image signs of as-yet undiagnosed, coexisting, subclinical conditions [99,100,101,102].

Extraction of radiomic image features beyond the scope of the human eye is expected to be another target for future projects. Different radiomic features can already be evaluated by the tool, building upon previous efforts from PyRadiomics [103]. Hereby, the infrastructure allows for AI-based large-scale image analysis of ill-defying radiomic features.

Recently, there have been similar efforts to provide an all-in-one AI-approach [104,105,106,107]. One example is AI-Rad Companion (Siemens Healthineers, Erlangen), an official medical product workflow solution interpreting CT thorax images [108]. Since the software is commercially available, every disease section has to be bought individually and the assessment is limited to ≤2 mm CT slice thickness. The company is constantly improving the algorithms and extending the addressed tasks, but it may not be extended by the customer themselves. Additionally, although the software provides a report of the findings, grading, risk assessment and therapy recommendations are not provided. Nevertheless, the costumer has the opportunity to purchase additional deep learning algorithms for specific regions of interests in MR and conventional X-ray, which our approach does not offer.

Another promising approach has been recently published by Pickhardt et al. at the RSNA (Radiological Society of North America) conference 2021 with an automated opportunistic screening possibility for cardiometabolic body composition biomarkers in CT abdomen [109]. Comparable to ours, the automated image assessment includes body tissue composition, osteoporosis and aortic calcium scoring. On the one hand, as expected, pulmonary evaluation is not targeted. On the other hand, osteoporosis detection already includes fracture evaluation. In addition to many parallels, a fusion of both tools with each implementing the more advanced algorithm could be achieved based on our developed workflow, since the technical infrastructure allows for the extension and up-scaling of additional and advanced algorithms, thereby enabling a persistent, dynamic improvement of the tool.

Here, a possible integration of healthcare data from digital documentation may be one of the future targets. As a perspective for the osteoporosis evaluation tool, it is planned to extend the network to a broader diagnostic assessment tool adding vertebra volume measurements, fracture grading, microarchitecture analysis and finally combining this information with the CT-based, automated BMD measurements. Additional extraction, validation and implementation of as-yet unidentified imaging biomarkers may also be targeted. Development of specified algorithms for lung parenchyma analysis concerning fibrotic changes are going to be future objectives as well as the identification of subtle imaging characteristics and biomarkers of malignancy risk prediction of pulmonary nodules. These characteristics are expected to have a significant diagnostic and prognostic value driving forward diagnosis at asymptomatic stage and consequently offering the opportunity for early therapy, potentially preventing disease manifestation. This development is expected to provide a profound clinical and scientific basis for improving patient care, risk assessment and disease therapy.

## 5. Conclusions

In Comorbidome, we aimed to develop an integral AI approach with the simultaneous automated detection, quantification and subsequent risk-classification of the most common pulmonary, metabolic, cardiovascular and musculoskeletal comorbidities in chest CT imaging. Generating an automated, holistic structured report allows a comprehensive overview of the complex medical situation of an individual patient in a preventive approach.

Moreover, this multidimensional tool provides an infrastructure to prospectively perform large-scale demographic studies on comorbidities of diseases of the chest, thereby contributing to sustainable improvement of patient care.

## Figures and Tables

**Figure 1 healthcare-10-02166-f001:**
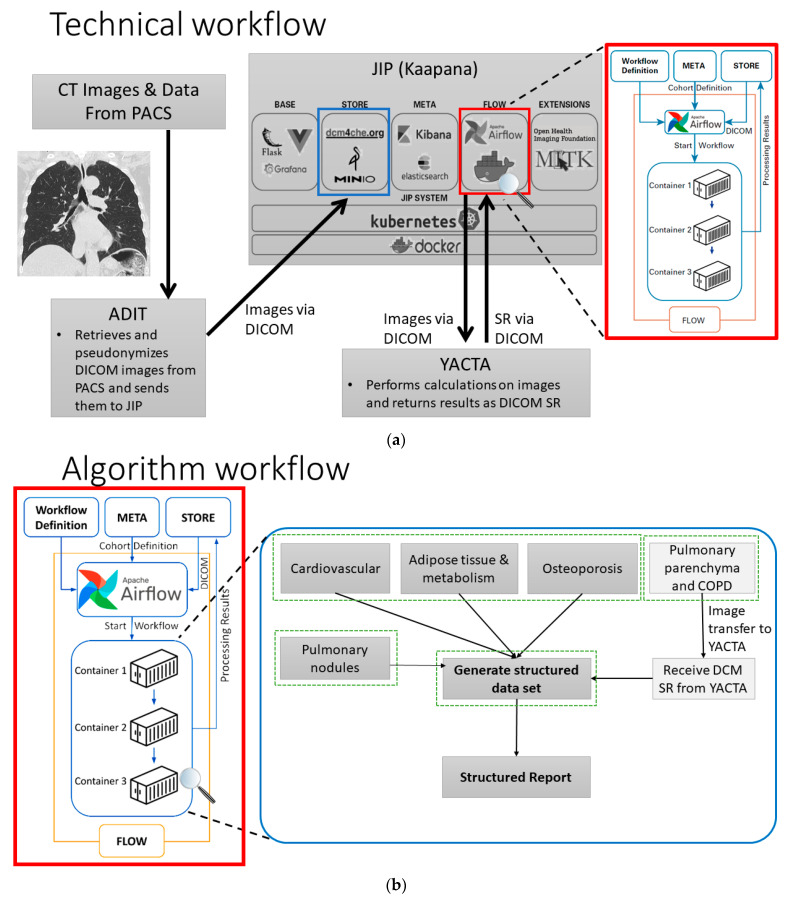
Flowchart of planned software integrations with: (**a**) Technical workflow and (**b**) Algorithm workflow.

**Figure 2 healthcare-10-02166-f002:**
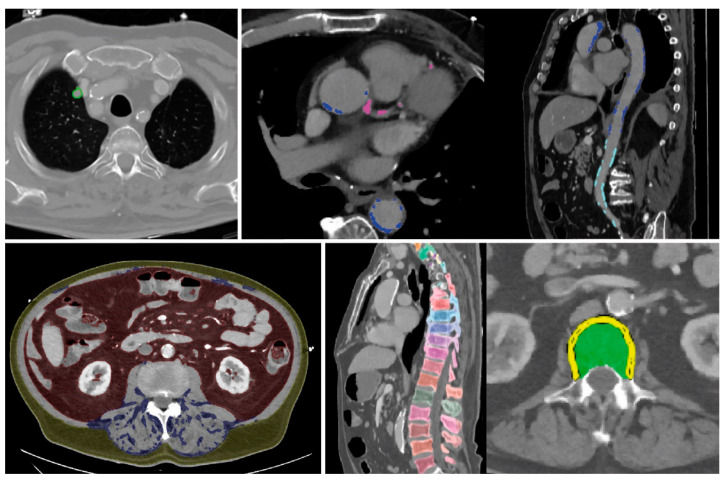
Example no. 1 ‘TAVI patient’—Summary of organ segmentations and pathological findings from left to right and top to bottom: pulmonary nodule of the right upper lobe, coronary and aortic calcifications, visceral, subcutaneous and intramuscular adipose tissue, and vertebral segmentations for assessment of BMD.

**Figure 3 healthcare-10-02166-f003:**
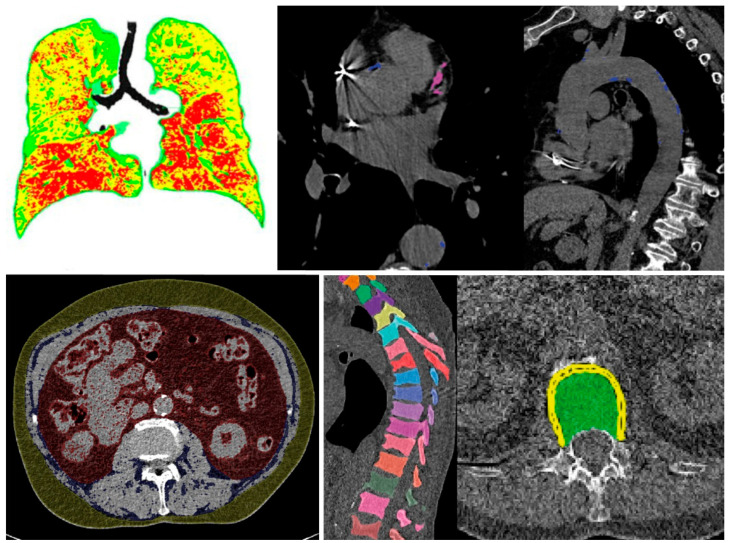
Example no. 2 ‘COPD patient’—Summary of organ segmentations and pathological findings from left to right and top to bottom: COPD, coronary and aortic calcifications, visceral, subcutaneous and intramuscular adipose tissue, and vertebral segmentations for assessment of BMD.

**Table 2 healthcare-10-02166-t002:** Summary of the results of Example No. 1 `TAVI patient’.

Comorbidity	Results	Conclusion
Adipose tissue and metabolism	− VFA 338 cm^2^ − SFA 148 cm^2^ − Intermuscular fat area (IMAT) 31 cm^2^ − Liver attenuation - − Liver fat content -	− Measurement of visceral and/or subcutaneous fat outside the range of standard deviation.
Pulmonary nodules	− Nodule right upper lobe − 13.48 mm^3^ − 1.6 mm × 3.5 mm × 4.6 mm	− Pulmonary Nodule with Lung-RADS 2 in the right upper lobe
Pulmonal parenchyma abnormalities (COPD)	− Emphysema Index 6% − Pi10 0.26 − Bronchiectasis Index 0.08%	− No Emphysema − Possible bronchial wall thickening.
Cardiovascular comorbidities	− CAC 399 − TAC 12284 − AAC 8214	− Moderate CAC. − High AAC and TAC with high risk for future cardiac events. Consider statin therapy, and high-intensity statin therapy if Agatston score 300.
Osteoporosis	− BMD 85 mg/cm^3^ (L1) − T-score −3.4 (L1)	− Suspicion of Osteoporosis with significantly reduced BMD

**Table 3 healthcare-10-02166-t003:** Summary of the results of Example No. 2 ‘COPD patient’.

Comorbidity	Results	Conclusion
Adipose tissue and metabolism	− VFA 235 cm^2^ − SFA 158 cm^2^ − Intermuscular fat area (IMAT) 31 cm^2^ − Liver attenuation 54HU − Liver fat content 7%	− Measurement of visceral and/or subcutaneous fat areas are normal. − Mild steatosis.
Pulmonary nodules	− No Nodule detected	
Pulmonal parenchyma abnormalities (COPD)	− Emphysema Index 33% − Pi10 0.21 − Bronchiectasis Index 0.52%	− Mild to moderate Emphysema − Possible signs of small airway disease type. − Possible bronchial wall thickening.
Cardiovascular comorbidities	− CAC 237 − TAC 2456 − AAC -	− Moderate CAC. − High TAC with high risk for future cardiac events. Consider statin therapy, and high-intensity statin therapy if Agatston score 300.
Osteoporosis	− BMD 52 mg/cm^3^ (L1) − T-score −4.6 (L1)	− Suspicion of Osteoporosis with significantly reduced BMD

## Data Availability

The datasets generated during and/or analyzed during the current study are available from the corresponding author on reasonable request.

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
