# Peer review of "AI-Supported Comprehensive Detection and Quantification of Biomarkers of Subclinical Widespread Diseases at Chest CT for Preventive Medicine"

_healthcare, 2022, doi:10.3390/healthcare10112166_

Round 1

Reviewer 1 Report

The use of nnU-net in the comprehensive detection of biomarkers for chest CT is quite interesting, however, there was already some similar research on craniomaxillofacial, parenchyma, and liver by using nnU-net to analyze CT scans.

Besides, some questions were raised:

  1. It is suggested to include a table of parameters of the example patient to highlight the correlation to the labeled results.
  2. What are the differences between the performance of the new-Net trained model, experts optimized model, and radiologists? 
  3. How long does it take to run a full analysis?
  4. Please also provide packages and software version information.

Reviewer 2 Report

The research focuses on highlighting AI algorithms and tools that simulate expert systems in artificial intelligence and rely on its algorithms directly in the medical field and in supporting early diagnosis of disease states for advanced ages and in diseases for which obesity is one of the main causes. The paper mainly focuses on supporting early diagnosis, but shows a number of aspects that need to be developed in the research paper, which can be summarized as follows:

§  A number of research references are somewhat outdated, although the field of AI has a strong scientific movement in the last five years, so I see the research references to be reviewed and updated for this study.

§  The various pathological aspects were reviewed with the algorithms narrating them significantly. It is preferable to focus on summarizing this part through an integrated mind map that includes the algorithms that were reviewed through research or through other similar research.

§  When presenting a number of useful algorithms are listed in this study, it is preferable to work on developing a vision for how to work on the initial diagnosis of diseases in the body, and thus choose the appropriate algorithm for treatment.

Reviewer 3 Report

AI-supported comprehensive detection and quantification of biomarkers of subclinical widespread diseases at chest CT for preventive medicine

The study basically presents the idea of evaluating more than one disease from a CT image of a patient by using Artificial Intelligence.
However, there are serious problems in the presentation of this study. For example, subheadings of Chapter 2 have switched between irrelevant topics (database presented in 2.2 and Cardiovascular Comorbidities presented in 2.5).
The presentation of the study is in a very complicated. The study need to be ordered as; Introduction, materials and methods, results, etc.
It is stated that deep learning is used but not detailed. What kind of structure does nnU-NET have, what are its advantages and disadvantages. Such issues should be specified, the parameters used should be presented.
Also, the adaptation of this nnU-NET for use in CT images should be clearly explained.
Image qualities are poor. The texts on image are hard to read and it is not possible to identify of symbols.
Detailed study is required for the publication of this study.

Round 2

Reviewer 3 Report

It has been observed that the answers and changes given to the criticisms made about deep learning consist of verbal guidance. I think that these additions are insufficient. The authors presented the results which were obtained by applying their data on deep learning. However, no information was provided on the structure of deep learning, how the data was adapted, and applied. This study needs more work to be more organized and explanatory in terms of deep learning. For this reason, I think that the study is not suitable for publication as it is.
